# Gender disparities in bladder cancer: A population-based study on life expectancy and health spending in Asia

Yi Hong Li[1], Yen-Chuan Ou[1]*, Min Che Tung[1]*, Yi Sheng Lin[1,2], Ya Chu Yang[3], Ying Ming Chiu[3,4,5], Chao Yu Hsu[1]

1 Division of Urology, Department of Surgery, Tungs' Taichung MetroHarbor Hospital, Taichung, Taiwan,
2 Doctoral Program in Translational Medicine, National Chung Hsing University, Taichung, Taiwan,
3 Department of Big Data Research Center, Tungs' Taichung MetroHarbor Hospital, Taichung, Taiwan,
4 Department of Allergy Immunology and Rheumatology, Tungs' Taichung MetroHarbor Hospital, Taichung, Taiwan, 5 Department of Post-Baccalaureate Medicine, College of Medicine, National Chung Hsing University, Taichung, Taiwan

* ycou228@gmail.com (YCO); tungminche@gmail.com (MCT)

## Abstract

### Background

The aim of this study was to elucidate the disparities in life expectancy, loss-of-life expectancy, and lifetime medical expenditure between sexes in patients with bladder cancer.

### Methods

In this retrospective study, we used three Taiwanese databases to analyze the data of patients diagnosed with bladder cancer between 2008 and 2019. Patients aged <30 years or >90 years were excluded. Survival and lifetime costs were estimated using the Kaplan–Meier and semiparametric methods. Subgroup analyses were performed to examine the effects of cancer stage, age, and factors such as hemodialysis on patient outcomes and costs.

### Results

This study included 30,390 new diagnoses of bladder cancer. Disparities in loss-of-life expectancy between men and women were observed in both non-muscle-invasive bladder cancer (3.17 [0.55] years for men vs. 7.14 [0.76] years for women) and muscle-invasive bladder cancer (8.86 [0.43] years for men vs. 10.64 [0.63] years for women). Carcinoma in situ revealed its profound impact, with the associated loss-of-life expectancy mirroring those of advanced stages (combined sex carcinoma in situ: 8.58 years, stage 2 men: 9.48 years, stage 2 women: 9.53 years). The cost per life-year showed a marked difference, especially for non-muscle-invasive bladder cancer ($4,631 for men vs. $7,636 for women) and muscle-invasive bladder cancer ($6,033

**Data availability statement:** This study is based on third-party data obtained from the National Health Insurance Research Database (NHIRD), which has been transferred to the Health and Welfare Data Science Center (HWDC), Ministry of Health and Welfare, Taiwan. The NHIRD contains de-identified healthcare data and is not publicly available due to government regulations. Access to the dataset requires formal application and approval from the HWDC. Interested researchers may apply for data access via the official website (https://dep.mohw.gov.tw/DOS/cp-5119-59201-113.html). The authors confirm that they do not have the right to share the raw data and that others can access the data in the same manner as the authors. The authors had no special privileges in obtaining the dataset. Proper citation and acknowledgment of the NHIRD as the data source are required when using the dataset.

**Funding:** The author(s) received no specific funding for this work.

**Competing interests:** The authors have declared that no competing interests exist.

**Abbreviations:** CIS, carcinoma in situ; CKD, chronic kidney disease; CPLY, Cost per life-year; ESRD, end-stage renal disease; LE, life expectancy; Loss-of-LE, Loss-of-life expectancy; MIBC, muscle-invasive bladder cancer; NHIRD, National Health Insurance Research Database; NMIBC, non-muscle-invasive bladder cancer; NTD, new Taiwan dollar; RC, radical cystectomy.

for men vs. $7,753 for women). Hemodialysis accounted for a significant portion of these costs, with hemodialysis rates of 4.6% in men and 18.5% in women.

## Conclusions

Women have a higher prevalence of high-grade histopathology and an extended duration of hemodialysis, culminating in inferior outcomes in non-muscle-invasive bladder cancer and muscle-invasive bladder cancer and augmented costs, compared with men. The role of hemodialysis and the carcinoma in situ stage highlights the need for vigilant monitoring and early aggressive treatment strategies.

## Introduction

Rapid advancements in bladder cancer treatments, including various chemotherapies and immunotherapies, incorporation of robotic surgery, and improvements in overall care quality, have presumably led to escalated associated costs. The prevalence of bladder cancer is significantly higher in men than in women, with a sex ratio of approximately 3:1–4:1 [1]; this difference has been attributed to higher smoking rates and exposure to occupational chemicals among men. However, this ratio is not mirrored in mortality rates. Most studies have reported higher mortality and morbidity rates in women than in men, associated with delayed diagnoses, later-stage presentations, longer referral times to urologists, and compromised treatment quality [2]. Some studies have similarly reported a higher incidence of nonurothelial cell carcinoma in women than in men [3,4]. Thus, extensive research has highlighted a noticeable sex disparity in the outcomes of bladder cancer treatments, with women often experiencing worse treatment outcomes than men [5]. Moreover, some studies suggest that factors such as hormones, gene mutations [6], and immunobiology may influence treatment responses, emphasizing the multifactorial nature of gender disparities in bladder cancer.

The exploration of these varied factors emphasizes the urgent need for more comprehensive and inclusive research methodologies to better understand the differences in bladder cancer prevalence and outcomes between the sexes. Therefore, the aim of this study was to estimate the life expectancy (LE), loss of LE, and lifetime medical expenditures of patients with bladder cancer after the disease diagnosis to evaluate the current gender disparities in Asian populations and identify vulnerable groups.

## Materials and methods

### Materials

This study was approved by the Institutional Review Board of Tungs' Taichung Metro Harbor Hospital (Taichung, Taiwan; approval number, 110034). This study was conducted in accordance with the ethical standards established in the 1964 Declaration of Helsinki and its later amendments. This study was conducted by Health and Welfare Data Center (Taipei City, Taiwan) and all individual information data were

de-identified. Informed consent could not be obtained from the study population. We used three databases provided by the Health and Welfare Data Science Center (Taipei, Taiwan), which are linked to each other. Data used in this study were accessed on 06/10/2021 for research purposes. First, we used the Taiwan Cancer Registry database to identify patients diagnosed with bladder cancer (International Classification of Diseases-O-3 code: C67) between 01/01/2008 and 31/12/2019. The authors did not have access to information that could identify individual participants during or after data collection. This database includes information on the cancer site, diagnosis date, pathological grade, muscle invasion, sex, and cancer stage. The National Health Insurance Research Database (NHIRD) was subsequently used to extract information on the medical expenditures and comorbidities of each patient; this database covers >99.6% of the Taiwanese population, representing a large source of data for nationwide research [7]. Finally, we used the National Mortality Registry data to confirm the survival status of the patients. Patients aged <30 years or >90 years were excluded. Specifically, patients aged 18–30 years were excluded from the study due to the small sample size of fewer than 100 cases, which would make statistical stratification unreliable and limit the feasibility of meaningful subgroup analysis. Additionally, patients aged over 90 years were excluded, as the average life expectancy in Taiwan is 76.94 years for men and 83.74 years for women [8]. These exclusions were deemed necessary to maintain the scientific rigor and interpretability of the findings.

## Statistical analysis

Each patient with bladder cancer was tracked from the date of cancer diagnosis until death or the end of the follow-up period, which was December 31, 2019. The survival function was generated using the Kaplan–Meier method until the end of the follow-up period. To extrapolate the survival function beyond the follow-up limit, the semiparametric survival extrapolation method proposed by Hwang et al. [9] was used. This method has been previously applied to various diseases, such as rheumatoid arthritis, psoriasis, stroke, and type 1 diabetes. The detailed steps are included in in Supplement S1 Text. To validate our extrapolation method, we compared life expectancy estimates extrapolated from the first 6 years with 12-year Kaplan–Meier estimates (S1 Table). In addition to the semiparametric extrapolation, we performed a multivariable Cox regression analysis to assess the independent effects of sex, age group, pathological grade, and cancer stage on overall survival. This model provided hazard ratios (HRs) with 95% confidence intervals and p-values for each factor, clarifying the statistical significance of individual components.

Hwang et al. [9] also proposed a method for estimating lifetime medical costs. Medical expenditure data were collected for each patient with bladder cancer after the cancer diagnosis. The monthly average healthcare cost was computed by summing the costs for all patients in a given month and dividing it by the number of surviving cases in that same month. Given the assumption that medical costs would increase before death, the average healthcare costs for the months preceding death were weighted to estimate the average cost function. The average cost function was subsequently multiplied by the corresponding monthly survival probability, and the costs for all months were summed to obtain lifetime medical costs. For consistency, the cost per life-year (CPLY) was derived by dividing the lifetime medical costs by the discounted LE (at a rate of 3% per year). All currency values were expressed in United Sates dollars (USD) in 2019 (1 USD = 30.93 New Taiwan dollars [NTD]). The R package iSQoL2 was used to estimate LE and lifetime costs.

## Results

A total of 30,390 new patients with bladder cancer diagnosed between 2008 and 2019 were identified, with men comprising 70% of the population. During the follow-up period, the composition of medical expenditures differed between males and females. In females, the distribution of costs was as follows: medications (14.9%), laboratory tests and imaging studies (34.1%), dialysis (39.7%), and medical consumables (4.4%). In contrast, the corresponding proportions in males were medications (19.2%), laboratory tests and imaging studies (52.0%), dialysis (16.1%), and medical consumables (4.4%). Table 1 and Fig 1 summarize LE, loss of LE, lifetime medical costs, and CPLY, all stratified, based on sex and age

**Table 1. Estimation of life expectancy, loss of life expectancy, lifetime cost (US) and cost per life-year for bladder cancer, stratified by sex and age at diagnosis.**

| sex | age | n | no. of death(%) | Age (mean±SD), years | LE (SE), years | Loss-of-LE (SE), years | Lifetime cost (SE), dollars[a] | Cost per life-year, dollars[a] |
|---|---|---|---|---|---|---|---|---|
| Male | | | | | | | | |
| | 30-59 | 4981 | 1115(22.39) | 52.29±6.37 | 21.60 (1.31) | 7.50 (1.30) | 81,563(4,032) | 4,489 |
| | 60-69 | 5645 | 1688(29.90) | 64.99±2.87 | 12.81 (0.46) | 6.20 (0.47) | 64,659(2,098) | 5,383 |
| | 70-79 | 6228 | 2928(47.01) | 75.12±2.86 | 8.27 (0.21) | 3.69 (0.21) | 43,412(1,064) | 5,399 |
| | 80-89 | 4438 | 2932(66.07) | 84.21±2.73 | 4.67 (0.11) | 2.35 (0.11) | 26,738(627) | 5,758 |
| Female | | | | | | | | |
| | 30-59 | 1801 | 460(25.54) | 52.86±6.24 | 19.99 (2.15) | 13.66 (2.17) | 141,602(10,171) | 8,308 |
| | 60-69 | 2403 | 748(31.13) | 65.17±2.86 | 12.35 (1.11) | 10.16 (1.11) | 95,821(5,512) | 8,233 |
| | 70-79 | 2983 | 1474(49.41) | 75.11±2.77 | 7.67 (0.26) | 6.72 (0.27) | 52,548(1,683) | 7,023 |
| | 80-89 | 1911 | 1319(69.02) | 84.06±2.66 | 4.07 (0.19) | 4.22 (0.18) | 25,500(1,136) | 6,312 |

SD, standard deviation. SE, standard error of mean.

[a]$1 dollar (US) = $30.93 dollars (New Taiwan).

at diagnosis. The data revealed that the younger the age group, regardless of sex, the greater was the loss of LE. Female patients with bladder cancer across all age groups had a lower LE than did their male counterparts. Additionally, women experienced a higher loss of LE than men, with nearly a twofold difference in every age group. When considering lifetime medical costs, women generally incurred higher costs than men, with the exception of the 80–89 age group. The CPLY for women peaked in the 30–59 age group ($8,308) and decreases with age, whereas for men, it peaked in the 80–89 age group ($5,758) and increased with age. The primary reason for the higher medical costs among women is the significantly higher proportion of women receiving long-term dialysis, compared with men. A more detailed explanation of this reason will follow in a later section.

The black lines represent sex-, age-, calendar year-matched referents simulated from the national vital statistics of Taiwan. The red dashed lines represent the real-world data from bladder cancer cohort and the blue dashed lines represent extrapolation from this bladder cancer cohort. The area between the black line and the red/blue dashed line represents the loss-of-life expectancy (loss-of-LE).

When the data were analyzed, based on the pathological grade of the disease, the loss of LE for patients with high-grade bladder cancer was approximately 2 years longer than that for patients with low-grade bladder cancer in men [mean (standard error of the mean): 6.12 (0.39) years vs. 4.28 (1.05) years] and in women [8.40 (0.52) years vs. 6.61 (1.48) years]. CPLY was higher in patients with high-grade bladder cancer than in patients with low-grade bladder cancer, with the highest cost observed in women with high-grade bladder cancer ($8,710) (Table 2).

Within the same cancer stage, women experienced a higher loss of LE than did men. This was especially notable in stage 1, where the difference in loss of LE between women and men was the largest [7.69 (1.00) years vs. 3.84 (0.77) years]. The CPLY for men gradually increased with advancing stages; however, for women, the CPLY remained similar from stages 0–3. Both sexes had the highest CPLY at stage 4 of the disease. Stage 0IS, also known as a flat tumor or carcinoma in situ (CIS), was separate from stage 0A. To ensure statistical reliability and minimize potential bias due to sample size imbalance, we combined both genders for analysis. Specifically, during stage stratification, there were 193 females and 341 males in the 0IS stage. Given the relatively small female sample size, combining both genders allowed for more reliable results and better accounted for significant differences between groups. Stage 0IS had a substantial impact on LE, with a loss of LE as high as 9.26 (1.00) years. This was significantly more than that of stage 0A and even more than that of the stage 1 and 2 groups (Table 3 and S1 Fig in the Supplement).

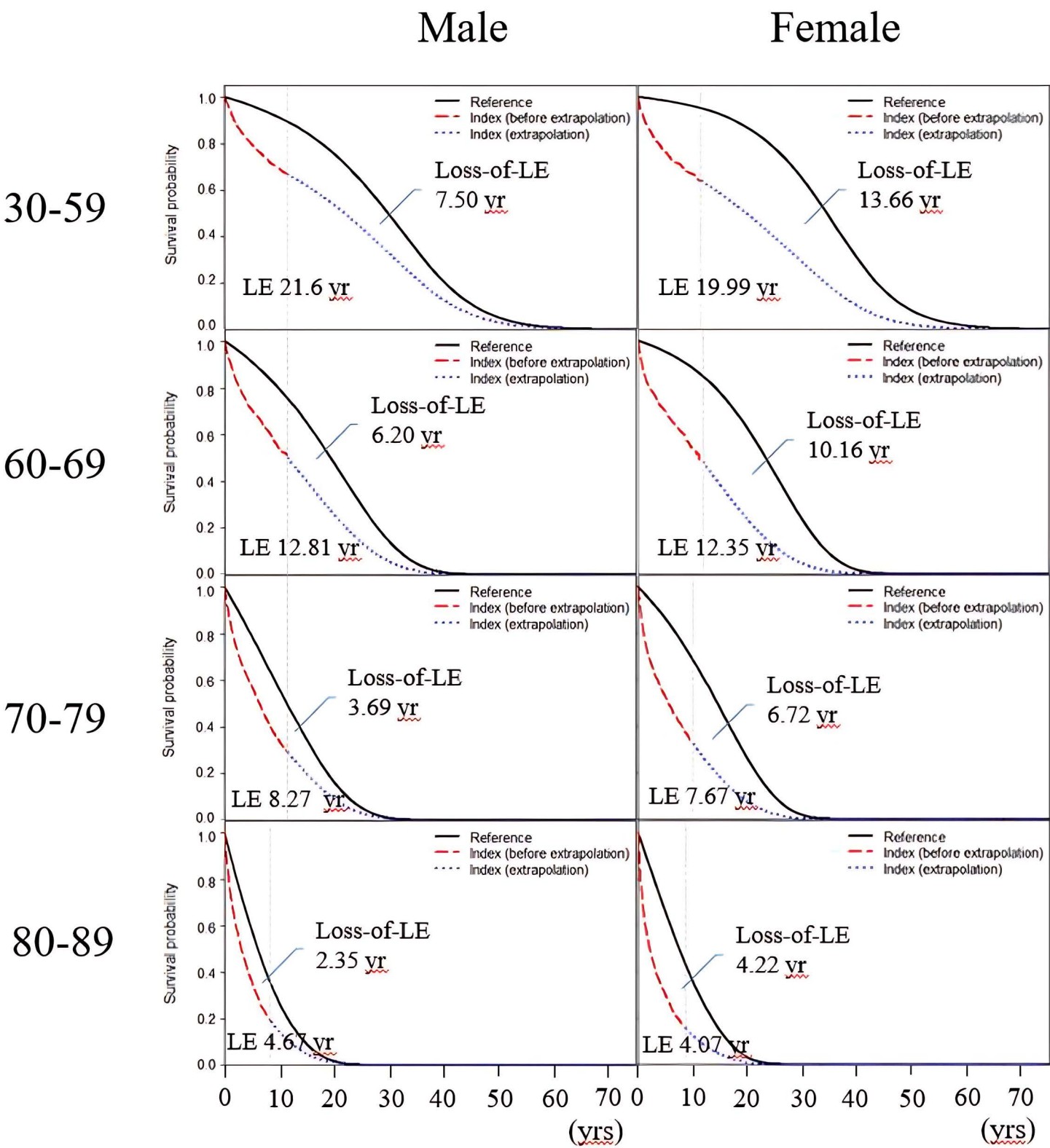

**Fig 1. Life expectancy (LE) and expected loss of life expectancy for bladder cancer, stratified by sex and age at diagnosis.**

**Table 2. Estimation of life expectancy, loss of life expectancy, lifetime cost (USD) and cost per life year for bladder cancer, stratified by sex and pathologic grading.**

| sex | grade | n | no. of death(%) | Age (mean±SD), years | LE (SE), years | Loss-of-LE (SE), years | Lifetime cost (SE), dollars[a] | Cost per life year, dollars[a] |
|---|---|---|---|---|---|---|---|---|
| Male | | | | | | | | |
| | Low | 6032 | 1651(27.37) | 67.42±12.65 | 13.64 (1.06) | 4.28 (1.05) | 54,756(2,709) | 4,316 |
| | High | 13231 | 5816(43.96) | 69.71±11.58 | 10.17 (0.37) | 6.12 (0.39) | 52,170(1,408) | 5,532 |
| Female | | | | | | | | |
| | Low | 2011 | 554(27.55) | 68.28±11.74 | 13.89 (1.49) | 6.61 (1.48) | 70,127(5,175) | 5,437 |
| | High | 5988 | 2783(46.48) | 70.51±10.99 | 10.23 (0.51) | 8.40 (0.52) | 81,487(2,920) | 8,710 |

SD, standard deviation. SE, standard error of mean.

[a]$1 dollar (US) = $30.93 dollars (New Taiwan).

**Table 3. Estimation of life expectancy, loss of life expectancy, lifetime cost (USD) and cost per life year for bladder cancer, stratified by sex and cancer stage.**

| sex | stage | n | no. of death (%) | no. of male (%) | Age (mean±SD), years | LE (SE), years | Loss-of-LE (SE), years | Lifetime cost (SE), dollars[a] | Cost per life year, dollars[a] |
|---|---|---|---|---|---|---|---|---|---|
| | 0IS | 534 | 184(34.46) | 341 (63.86) | 68.91±10.67 | 8.58 (0.88) | 9.26 (1.00) | 60,275(5,592) | 7,132 |
| Male | | | | | | | | | |
| | 0A | 5303 | 1238(23.35) | | 67.79±12.35 | 15.08 (1.09) | 2.61 (1.09) | 62,769(4,763) | 4,594 |
| | 1 | 6674 | 2122(31.8) | | 68.75±11.85 | 13.09 (0.79) | 3.84 (0.77) | 55,394(1,979) | 4,613 |
| | 2 | 2972 | 1442(48.52) | | 69.88±11.86 | 9.48 (0.72) | 6.70 (0.73) | 47,565(2,182) | 5,394 |
| | 3 | 1446 | 885(61.2) | | 70.63±11.40 | 6.95 (0.59) | 8.69 (0.62) | 40,282(1,775) | 6,239 |
| | 4 | 1379 | 1174(85.13) | | 70.30±11.86 | 2.87 (0.48) | 13.00 (0.50) | 27,305(1,741) | 9,935 |
| Female | | | | | | | | | |
| | 0A | 2123 | 518(24.4) | | 68.22±11.14 | 14.10 (1.58) | 6.37 (1.60) | 93,090(7,595) | 7,131 |
| | 1 | 2769 | 979(35.36) | | 69.50±11.08 | 11.74 (0.96) | 7.69 (1.00) | 86,985(5,267) | 7,952 |
| | 2 | 1340 | 688(51.34) | | 71.85±10.88 | 9.53 (0.84) | 8.06 (0.88) | 66,565(4,282) | 7,658 |
| | 3 | 601 | 391(65.06) | | 72.88±11.45 | 6.05 (0.97) | 10.79 (1.05) | 44,086(4,549) | 7,708 |
| | 4 | 728 | 640(87.91) | | 71.83±11.36 | 2.55 (0.61) | 15.06 (0.70) | 25,050(2,680) | 10,452 |

SD, standard deviation. SE, standard error of mean.

[a]$1 dollar (US) = $30.93 dollars (New Taiwan).

Based on whether the cancer was muscle-invasive, the cancer stage was divided into stages 0–1 and stages 2–4. The difference in loss of LE between stages 0–1 and stages 2–4 was greater in women [7.14 (0.76) years vs. 10.64 (0.63) years] than in men [3.17 (0.55) years vs. 8.86 (0.43) years]. Furthermore, the difference in CPLY was higher in women (stages 0–1 and stages 2–4: $7,636 vs. $7,753) than in men ($4,631 vs. $6,033) (Table 4). We subsequently examined the relationship between cancer stage and age groups. In females, MIBC was more prevalent than in males for those aged above 70, whereas NMIBC was more common in females than in males for those aged below 70 (S2 Fig).

To validate our extrapolation method, we compared life expectancy estimates extrapolated from the first 6 years with 12-year Kaplan–Meier estimates (S1 Table). The relative bias was minimal, ranging from –6.85% to 1.85% in males and from –2.14% to 1.80% in females, demonstrating good agreement between extrapolated and observed estimates. These findings support the robustness of our approach despite inherent uncertainties. To elucidate prognostic factors, we

**Table 4. Estimation of life expectancy, loss of life expectancy, lifetime cost (USD) and cost per life year for bladder cancer, stratified by sex and cancer stage.**

| sex | stage | n | no. of death(%) | Age (mean±SD), years | LE (SE), years | Loss-of-LE (SE), years | Lifetime cost (SE), dollars[a] | Cost per life-year, dollars[a] |
|---|---|---|---|---|---|---|---|---|
| Male | | | | | | | | |
| | 0-1 | 12318 | 3481(28.26) | 68.33±12.06 | 14.08 (0.55) | 3.17 (0.55) | 59,269(2,308) | 4,631 |
| | 2-4 | 5797 | 3501(60.39) | 70.16±11.75 | 7.16 (0.40) | 8.86 (0.43) | 40,468(1,314) | 6,033 |
| Female | | | | | | | | |
| | 0-1 | 5085 | 1560(30.68) | 68.96±11.08 | 12.73 (0.75) | 7.14 (0.76) | 90,242(3,887) | 7,636 |
| | 2-4 | 2669 | 1719(64.41) | 72.08±11.15 | 6.79 (0.63) | 10.64 (0.63) | 48,533(2,731) | 7,753 |

SD, standard deviation. SE, standard error of mean.

[a]$1 dollar (US) = $30.93 dollars (New Taiwan).

conducted a multivariable Cox regression including sex, age, tumor grade, and cancer stage (S2 Table). After adjustment, male sex was associated with lower mortality (HR 0.90, 95% CI 0.87–0.94, p<0.01). Compared to patients aged 30–59, those aged 60–69, 70–79, and 80–89 had HRs of 1.49, 2.50, and 4.62, respectively (all p<0.01). High-grade tumors (HR 1.52) and advanced-stage disease (HR 2.72) also significantly increased mortality risk (p<0.01).

We analyzed the cost components to understand the variability in the CPLY. The primary driver of higher medical costs among women was their significantly greater proportion of undergoing dialysis, compared with men, with a ratio of 22.02% to 7.39%. A subgroup analysis examining the timing of hemodialysis before and after a bladder cancer diagnosis showed that a larger percentage of women had already been on hemodialysis (84.1%), compared with men (62.7%). Furthermore, in the female cohort, younger patients comprised the majority of patients on dialysis before being diagnosed with bladder cancer, with 90.71% aged 30–59 years. This trend was not as evident in the male cohort, in which approximately 60% of all age groups were on dialysis.

To better understand the cost burden of dialysis, we analyzed the actual dialysis costs and total medical costs for both dialysis and non-dialysis patients during follow-up. We calculated the proportion of dialysis costs relative to total medical costs by summing the dialysis expenses for all dialysis patients and dividing this total by the overall medical costs for all patients. Unlike the CPLY, which was estimated using extrapolation, this analysis presents observed dialysis costs as they occurred in real-time.

Our findings indicate that female patients in the 30–59 age group accounted for the highest proportion of dialysis costs, with this age group contributing 49.98% of dialysis costs relative to total medical costs. This proportion decreased steadily with increasing age, reaching 19.53% in the 80–89 age group. A similar downward trend was observed in male patients; however, the proportions were significantly lower. Among men, dialysis costs constituted 20.75% of total medical expenses in the 30–59 age group, decreasing to 8.26% in the 80–89 age group. Across all age groups, dialysis accounted for 39.68% of total costs in females compared to just 16.1% in males.

Additionally, women with different disease stages showed a prolonged duration of hemodialysis. In both sexes, patients with more advanced disease stages underwent hemodialysis for a longer period than that of their counterparts (Table 5 and Fig 2).

The x-axis represents the timeline of initiating hemodialysis, with 0 denoting the time of bladder cancer diagnosis. Negative values indicate hemodialysis initiation before the bladder cancer diagnosis, while positive values indicate hemodialysis initiation after the bladder cancer diagnosis. For instance, 1 indicates hemodialysis initiation within 1 year after the bladder cancer diagnosis, and >5 indicates hemodialysis initiation more than 5 years after the bladder cancer diagnosis.

**Table 5. Number of bladder cancer patients receiving long-term dialysis, stratified by sex, age at diagnosis, and proportion of dialysis costs during follow-up.**

|  | age | Number of Patients (n) | Dialysis Patients (n, %) | Dialysis Before Dx (n, %) | Dialysis After Dx (n, %) | Dialysis costs/ Total medical costs (%) during follow-up (million USD) |
|---|---|---|---|---|---|---|
| Male |  |  |  |  |  |  |
|  | 30-59 | 4981 | 385(7.73) | 245(63.64) | 140(36.36) | 27.62/133.06(20.75) |
|  | 60-69 | 5645 | 510(9.03) | 322(63.14) | 188(36.86) | 28.82/150.65(19.13) |
|  | 70-79 | 6228 | 484(7.77) | 294(60.74) | 190(39.26) | 22.76/164.4(13.85) |
|  | 80-89 | 4438 | 194(4.37) | 126(64.95) | 68(35.05) | 7.43/89.98(8.26) |
|  | All | 21292 | 1573(7.39) | 987(62.75) | 586(37.25) | 86.63/538.1(16.1) |
| Female |  |  |  |  |  |  |
|  | 30-59 | 1801 | 538(29.87) | 488(90.71) | 50(9.29) | 43.52/87.08(49.98) |
|  | 60-69 | 2403 | 664(27.63) | 585(88.10) | 79(11.90) | 44.49/98.8(45.03) |
|  | 70-79 | 2983 | 603(20.21) | 462(76.62) | 141(23.38) | 30.67/93.85(32.68) |
|  | 80-89 | 1911 | 198(10.36) | 150(75.76) | 48(24.24) | 7.45/38.15(19.53) |
|  | All | 9098 | 2003(22.02) | 1685(84.12) | 318(15.88) | 126.13/317.89(39.68) |

**Fig 2. The cumulative proportion of hemodialysis initiation in bladder cancer patients over time, stratified by sex and stage.**

## Discussion

Over the past decade, the sex ratio for bladder cancer diagnosis in Taiwan has been 2.4:1 [10]. This ratio is notably lower than the ratios of 3:1–4:1 frequently cited in previous studies. Sex disparities in bladder cancer outcomes have been extensively researched and can be attributed to various factors, including smoking habits, hormonal influences, delays in diagnosis upon initial presentation, postponed referral to urologists, and differences in adherence to clinical guidelines [11]. Women are often diagnosed at more advanced stages of the disease, which further exacerbates this disparity. Nevertheless, in our study, the proportion of women was marginally higher than that of men (34.4% vs. 32.0%).

### Cancer stage: non-muscle-invasive bladder cancer

In both sexes, a significant proportion of patients are diagnosed at stages Tis to T1, and are classified as presenting non-muscle-invasive bladder cancer (NMIBC). A substantial body of the literature posits that within this NMIBC cohort, women generally have worse prognosis than men. Our findings corroborate this assertion because female patients have more than twice as much loss of LE as their male counterparts (7.14 years vs. 3.17 years). Some investigations have highlighted that women often present with high-grade disease on histopathological examinations [11]. Consistent with this finding, our study revealed the predominant presence of high-grade disease in women. Moreover, meta-analyses that did not separate data by grade (Tis to T1) reported higher recurrence rates, poorer responses to *Bacillus Calmette-Guerin* (BCG) therapy, and higher cancer-specific mortality in women than in men [12].

However, it is essential to recognize the existence of contradictory findings. Some studies have contended that sex discrepancies in NMIBC are negligible but become prominent in muscle-invasive bladder cancer (MIBC) [13]. Adding to this discourse, one study posited that even minor delays in treating NMIBC could escalate the risks of disease progression and recurrence [14]. This finding complements the existing literature that underscores the detrimental effects of treatment decisions, particularly in women.

CIS, also referred to as Tis in our study and categorized as stage 0, often has a less favorable prognosis, particularly in patients who do not respond well to BCG treatment [15]. CIS has a pronounced proclivity for the recurrence and progression of MIBC. The standard treatment for CIS involves transurethral resection of the bladder tumor, followed by intravesical BCG therapy. For patients unresponsive to this initial line of treatment, more intensive alternatives, such as radical cystectomy (RC) or bladder-sparing therapy, are advised. Given their complexity, these interventions naturally result in heightened loss of LE and associated costs.

Although most studies have focused on concurrent (or concomitant) CIS, patients with primary CIS appear to have worse outcomes [16], a finding specifically addressed in our study. The resultant LE and costs were comparable to those observed for stage 2 bladder cancer (LE for combined CIS is 8.58 years, whereas it is 9.48 years for men with stage 2 and 9.53 years for women with stage 2). Hence, our findings contradict those of previous studies and draw parallels between CIS and other categorizations, underscoring the need for vigilant monitoring and aggressive treatment of patients with CIS.

In our subgroup analysis, which examined the interplay between cancer stage and age, we observed a notable increase in aggressive stage diagnoses among female patients aged >70 years. Male patients aged <70 years were conversely more likely to be diagnosed with MIBC at the initial presentation. This finding challenges the widely held belief that a late diagnosis is solely responsible for more advanced diseases in women. We propose that factors such as improved clinical access, employer-sponsored health checks, and greater health awareness, particularly among younger individuals, may have had a role in this shift in the epidemiological pattern.

### Cancer stage: muscle-invasive bladder cancer

RC is the standard treatment of choice for resectable MIBC. The bulk of the literature covers post-RC treatment efficacy and the responses to neoadjuvant and adjuvant chemotherapies. Of note, a meta-analysis consistently suggested that female individuals had worse cancer-specific survival (CSS), overall survival (OS), and disease-free survival than did male

individuals [17]. An interesting finding is that, although the meta-analysis included 10 studies centered on Asian populations, no significant differences were found in the subgroup analyses of CSS, OS, and disease-free survival. Upon closer inspection, these 10 studies primarily involved populations from Korea and Japan, with some being compromised in terms of quality, particularly in the analysis of OS. Given this context, we believe that our study provides valuable insights, particularly by addressing gaps in Asian demographics and offering new perspectives.

Our data indicated that, although women experience a more pronounced loss of LE than did men, the disparity is less pronounced than in MIBC. In particular, women faced a loss of LE of 10.64 years, compared with 8.86 years for men. Upon age stratification, we detected an encouraging trend toward a reduced MIBC incidence in younger women. However, these slight variations do not fully explain the increased loss of LE in the female demographic. We unfortunately could not quantify the percentage of patients who underwent RC or their subsequent chemotherapy responses.

The rationale behind sex disparities in post-RC patient outcomes remains a topic of considerable debate. Numerous meta-analyses have indicated that women generally have poorer CSS, OS, and recurrence rates than do men [18]. Although some reports have attributed this factor to women often being diagnosed at a more advanced stage or encountering diagnostic delays, the reasons remain speculative. A potential contributing factor can be differing treatment patterns, with men being more likely than women to receive curative treatment [19]. However, in contrast to these findings, another study found no significant sex differences in treatment quality [20], although women tended to start treatment earlier than men. On a different note, a meta-analysis explored the diminishing gap, positing that women receiving neoadjuvant chemotherapy coupled with RC often experienced superior results in terms of recurrence and CSS, although there was no discernible difference between RC with adjuvant chemotherapy [21]. A similar trend was observed in another observational study, further supporting these findings [22].

**Cost per life-year**

Among women, although the average CPLY was significantly higher than that for men, the costs for stages Ta to 3 remained relatively consistent, with only stage 4 incurring the highest costs for both sexes. This finding differs from that of a population study in Germany [23], which showed that high-risk NMIBC incurred the highest cumulative follow-up costs, unlike MIBC. In that study, follow-up using computed tomography urography was the most costly, followed by cystoscopy.

While previous studies with similar designs have consistently shown an increasing trend in costs with age for both males and females [24,25], our study reveals a contrasting pattern in females. Specifically, male patients in our study follow the expected trend of rising CPLY with age, whereas female patients demonstrate an opposite pattern. Examining the breakdown of costs within Taiwan's National Health Insurance, we found that hemodialysis dominated expenses for both sexes across all age groups. Further analysis of the proportion of dialysis costs relative to total medical expenses indicated a similar trend in both genders, with the highest costs observed in the 30–59 age group. However, the proportion of all age groups was significantly lower in males (16.1%) compared to females (39.68%), highlighting a notable gender disparity in dialysis-related financial burden.

The substantial cost of established hemodialysis significantly influences overall expenditure, particularly for women in the 30–59 age group. The higher proportion of dialysis costs in women may be attributed to several factors, including a greater prevalence of long-term hemodialysis and a higher likelihood of starting dialysis before bladder cancer diagnosis (84.1% in females vs. 62.7% in males). Younger women in particular (90.71% aged 30–59 years) were already on dialysis prior to their cancer diagnosis. This cumulative cost of long-term dialysis could explain the pronounced impact on their CPLY.

Although our data did not directly indicate this preference, our clinical experience suggests that Taiwanese patients frequently opt for trimodal therapy over RC. One study [26] explored the cost difference between trimodal therapy and RC, revealing that the former had higher expenses within a 1-year follow-up period. To determine which treatment is more cost-intensive and understand the potential influence of sex differences, a more extensive investigation over an extended timeframe may be required.

 

## Influence of hemodialysis

Hemodialysis has profound implications for patient survival, comorbidities, and financial burden. It appears to be associated with a higher mortality rate than that of certain cancers [27]. Our data revealed that patients with advanced disease had a greater prevalence of hemodialysis. We propose that hemodialysis suggests an elevated risk of developing bladder cancer and indicates progression to more advanced stages. A study from Taiwan demonstrated that patients with chronic kidney disease (CKD) on dialysis were at a heightened risk of genitourinary cancers, with bladder cancer being notably prevalent [28]. Of interest, women had a higher hazard ratio than did men (13.94 vs. 4.9), with younger age groups being particularly at risk. These findings align with our observation of higher dialysis rates among women and suggest that CKD and prolonged hemodialysis may disproportionately affect female patients. Numerous studies have scrutinized the impact of CKD on oncological outcomes after RC [29,30]. These studies have uniformly reported reduced cancer-specific and OS rates. Another study identified adverse pathological features along with increased rates of transfusion and readmission [30]. In patients with end-stage renal disease, undergoing RC is associated with substantial complications and mortality [31]. Moreover, patients with CKD continue to exhibit an elevated incidence of complications when transurethral resection of the bladder tumor is performed for NMIBC [32].

Prolonged hemodialysis also poses risks such as cardiovascular diseases, infections, and other morbidities. This logically accounts for the higher costs associated with younger patients who, despite having comparable stage distributions, bear more expenses and experience greater loss of LE due to the predominant prevalence of hemodialysis. Additionally, although cisplatin-eligible patients generally experience better therapeutic outcomes, patients on dialysis or who have a glomerular filtration rate <60 mL/min are typically excluded from standard treatment protocols. This exclusion may explain the higher dialysis rates among female patients because a smaller proportion is eligible for standard therapies.

To the best of our knowledge, this study is the first to provide stage-specific stratification of bladder cancer outcomes. Sex disparities in these outcomes can be attributed to various factors. Two recent population studies from Norway and the Netherlands showed that although female patients with bladder cancer had higher mortality rates at diagnosis than did their male counterparts, this pattern shifted in favor after 2 years [33,34]. This trend suggests an indirect possibility that women may be diagnosed at more advanced disease stages, evidenced by a male-to-female ratio in MIBC of 29%:31% (p<0.001), because of potential delays in urologist referrals. Our findings indicated a notably higher prevalence of MIBC in female patients than in male patients, with a male-to-female ratio of 32% to 34.4%. Within the Tis stage, women initially had worse outcomes did than men, but by the 6th year, this trend was reversed in favor of women (S3 Fig). However, in all other stages, women consistently had worse outcomes than that of men.

## Clinical implications for high-risk populations

We introduce a novel concept regarding the impact of hemodialysis on the diagnosis of bladder cancer. Young female patients undergoing hemodialysis often face unique diagnostic challenges owing to anuria or oliguria. These patients are at a particularly high risk of poor bladder cancer outcomes, which raises concerns about the adequacy of current diagnostic and screening protocols. Although routine cystoscopy for bladder cancer screening is not widely recommended in the general population, we propose that early detection strategies may be beneficial for this high-risk group. Targeted screening, such as periodic cystoscopy, may improve outcomes by enabling earlier detection and treatment in young females on hemodialysis, in whom diagnosis is frequently delayed owing to their clinical presentation.

## Limitations

This study has some limitations. We were unable to fully account for various factors, including smoking, occupational exposure to chemicals, underlying diseases, socioeconomic status, and insurance status, all of which may influence mortality outcomes. However, we attempted to minimize these differences by matching the study population with the general population based on age, sex, and year of diagnosis. Additionally, the Monte Carlo-based reference group may

not capture key factors (e.g., comorbidities, socioeconomic status), potentially biasing comparisons. Moreover, our model assumes patient homogeneity for the included variables. In reality, variations in treatment, healthcare access, and other factors may introduce residual confounding. Regarding medical costs, self-paid medications and caregiving expenses are not covered by Taiwan's National Health Insurance, were not included in our cost analyses. This omission could underestimate the financial burden of the disease on individuals and their families. With respect to race, as the National Health Insurance covers >99.9% of the Taiwanese population, the representation of white, black, and other racial groups is minimal. Some studies have reported similar mortality rates between Caucasian and Asian populations, whereas black patients tend to have worse outcomes, possibly due to more advanced stages and higher-grade disease at diagnosis [35]. Additionally, reliance on database records poses limitations, such as the potential for incomplete or outdated data, errors in data entry or collection, and biases, because the data may not be fully representative of the entire population. These factors may affect the comprehensiveness and accuracy of our findings. Nonetheless, we have taken stringent measures to validate our data and have openly discussed these limitations to ensure transparency and maintain the integrity of our study conclusions.

## Conclusions

Our study examines sex disparities in bladder cancer outcomes, highlighting the marked differences in medical implications and costs. In particular, younger women face heightened risks with an increased loss of LE and costs. Contrary to prior assumptions, delayed diagnoses are not the primary reason for advanced disease in women, especially because younger women have fewer advanced cases. However, women have worse outcomes in both NMIBC and MIBC, likely driven by high-grade histopathology and prolonged hemodialysis. Aggressive follow-up and early surgical interventions, especially for CIS, are crucial, because CIS outcomes often resemble those of more advanced stages. A pivotal finding is the under-recognized effect of hemodialysis on outcomes, which is associated with higher comorbidity and limited treatment options. Our study underscores the need for prompt and accurate diagnosis, especially in women, and suggests improvements in treatment approaches. Despite some data omissions, this study offers key insights for healthcare professionals and stresses the importance of addressing sex disparities in bladder cancer outcomes.

## Supporting information

**S1 Text. The semiparametric survival extrapolation method used in this study involves three main steps.** (DOCX)

**S1 Table. Validation of extrapolated life expectancy estimates in bladder cancer patients: comparison of 6-year follow-up extrapolations with 12-year Kaplan–Meier estimates.**[a] (DOCX)

**S2 Table. Multivariable cox proportional hazards regression analysis of overall survival in bladder cancer patients.** (DOCX)

**S1 Fig. Comparison of life expectancy (LE) loss estimates across multiple cohorts.** For female bladder cancer patients diagnosed at stage 1, their life expectancy after diagnosis is 11.74 years, which is 1.35 years shorter than that of male bladder cancer patients diagnosed at the same stage (whose life expectancy after diagnosis is 13.09 years). Life expectancy loss refers to the difference between the life expectancy in the study cohort and that of a reference population matched by age, sex, and calendar year, simulated using life tables. A comparison of life expectancy loss, or the

difference in life expectancy loss (7.69 − 3.84 = 3.85), represents a difference-in-differences adjusted for potential confounding factors. Values are presented as mean ± SEM.
(DOCX)

**S2 Fig. Stacked bar chart illustrates the distribution of NMIBC, MIBC across age groups for males and females.** Both genders exhibited a similar trend, with the percentage of NMIBC decreasing and the percentage of MIBC increasing in older age groups. The proportion of cases classified as unknown remained consistent across all age groups and genders. In age groups below 70, females demonstrated a comparable or slightly higher percentage of NMIBC compared to males, whereas in age groups above 70, the percentage of NMIBC was significantly lower in females. Conversely, the percentage of MIBC showed an opposite pattern, with higher proportions observed in females in the older age groups.
(JPG)

**S3 Fig. Long term survival after bladder cancer diagnosis, stratified by sex and stage.**
(DOCX)

## Acknowledgments

Not applicable.

## Author contributions

**Conceptualization:** Yi Hong Li, Yen-Chuan Ou.

**Data curation:** Yi Sheng Lin.

**Formal analysis:** Ya Chu Yang.

**Methodology:** Ying Ming Chiu.

**Software:** Ya Chu Yang, Ying Ming Chiu.

**Supervision:** Min Che Tung, Chao Yu Hsu.

**Writing – original draft:** Yi Hong Li.

**Writing – review & editing:** Yen-Chuan Ou, Min Che Tung, Yi Sheng Lin, Ying Ming Chiu.

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
