## [Decision Letter · Decision Letter 0]

5 Jan 2025

PONE-D-24-48193Gender disparities in bladder cancer: A population-based study on life expectancy and health spending in AsiaPLOS ONE

Dear Dr. Ou,

Thank you for submitting your manuscript to PLOS ONE. After careful consideration, we feel that it has merit but does not fully meet PLOS ONE’s publication criteria as it currently stands. Therefore, we invite you to submit a revised version of the manuscript that addresses the points raised during the review process.

We look forward to receiving your revised manuscript.

Kind regards,

Mazyar Zahir, MD

Academic Editor

PLOS ONE

Journal Requirements:

For additional information about PLOS ONE ethical requirements for human subjects research, please refer to http://journals.plos.org/plosone/s/submission-guidelines#loc-human-subjects-research .

Reviewers' comments:

Reviewer's Responses to Questions

**Comments to the Author**

1. Is the manuscript technically sound, and do the data support the conclusions?

Reviewer #1: Yes

Reviewer #2: Yes

2. Has the statistical analysis been performed appropriately and rigorously? 

Reviewer #1: Yes

Reviewer #2: I Don't Know

3. Have the authors made all data underlying the findings in their manuscript fully available?

Reviewer #1: Yes

Reviewer #2: No

4. Is the manuscript presented in an intelligible fashion and written in standard English?

Reviewer #1: Yes

Reviewer #2: Yes

5. Review Comments to the Author

Reviewer #1: Dear Author

Thank you for your retrospective cross-sectional study.

1: Was all the information you needed available in the patient file?

2: Why did you exclude all patients under the age of 30 from the study?

Reviewer #2: Thank you for the opportunity to review this manuscript, and congratulations to the authors for conducting this study. The study from Taiwan, based on database exploration, compared life expectancy and medical expenditure between males and females, revealing interesting findings in their population.

Here are some major issues worth addressing to improve the presentation of the work:

1. The reason for excluding individuals under 30 and over 90 years of age from the study should be added.

2. Some issues in the discussion need further elaboration. For example, the following finding requires more explanation: The CPLY for women peaked in the 30–59 age group ($8,308) and decreased with age, whereas for men, it peaked in the 80–89 age group ($5,758) and increased with age.

3. Why didn’t you divide males and females with 0IS in Table 3A? Throughout the manuscript, 0IS was not stratified by sex.

4. The primary driver of higher medical costs was introduced as hemodialysis. However, there is no information on how much of the costs were due to dialysis and the role of other expenditures. Providing this information would clarify it for the readers.

5. In the discussion section, you mentioned, “In our subgroup analysis, which examined the interplay between cancer stage and age, we observed a notable increase in aggressive stage diagnoses among female patients aged >70 years.” More clarification is suggested regarding the interplay between cancer stage and age for each sex in the results section.

Additionally, here are some minor issues suggested for improvement:

1. In the results section of the abstract, it seems that “male” and “female” have been used incorrectly in this phrase: (3.17 [0.55] years for women vs. 7.14 [0.76] years for men).

2. Using MeSH terms is recommended for keywords.

3. Please recheck the numbers in the tables. For example, the percentages of the number of deaths in Table 3 seem incorrect.

4. In the discussion section, you mentioned, “CIS, also referred to as Tis in our study and categorized as stage 1, often…”. Doesn’t CIS categorize as stage 0?

6. PLOS authors have the option to publish the peer review history of their article (what does this mean? ). If published, this will include your full peer review and any attached files.

**Do you want your identity to be public for this peer review?** For information about this choice, including consent withdrawal, please see our Privacy Policy .

Reviewer #1: No

Reviewer #2: No

---

## [Author Response · Author response to Decision Letter 1]

6 Feb 2025

Reviewer #1:

Dear Author

Thank you for your retrospective cross-sectional study.

1: Was all the information you needed available in the patient file?

Answer:

Thank you for your question. As noted in the "Limitation" section of the manuscript, certain factors—such as smoking habits, occupational exposure to chemicals, socioeconomic status, and insurance status—could not be directly obtained due to the inherent constraints of retrospective studies and database-based research. Additionally, self-paid medications and caregiving expenses, which are not covered by Taiwan’s National Health Insurance, could not be included in the analysis. While these limitations exist, we mitigated potential biases by matching our study population with the general population based on age, sex, and year of diagnosis. This limitation has been explicitly discussed in the manuscript to provide transparency regarding the scope and boundaries of our findings (page 29-30, line 459 to 477 in tracked changes file; page 28-29, line 445 to 463 in without tracked changes file). We also added an explanation in the text that such results might overlook the actual economic burden that patients and their families have to bear.

2: Why did you exclude all patients under the age of 30 from the study?

Answer:

Thank you for your comment. Patients aged <30 years was excluded for the following reasons: Patients aged 18–30 years were excluded because the number of cases in this age group was fewer than 100, making statistical stratification unreliable and limiting the potential for meaningful subgroup analysis. This exclusion was necessary to ensure the scientific rigor and interpretability of the study findings. This explanation has been added to the "Materials and Methods" section of the manuscript (page 7, line 104 to 110 in tracked changes file; page 7, line 103 to 109 in without tracked changes file).

Reviewer #2:

Thank you for the opportunity to review this manuscript, and congratulations to the authors for conducting this study. The study from Taiwan, based on database exploration, compared life expectancy and medical expenditure between males and females, revealing interesting findings in their population.

Here are some major issues worth addressing to improve the presentation of the work:

1. The reason for excluding individuals under 30 and over 90 years of age from the study should be added.

Answer:

Thank you for your comment. Patients aged <30 years or >90 years were excluded for the following reasons: Patients aged 18–30 years were excluded because the number of cases in this age group was fewer than 100, making statistical stratification unreliable and limiting the potential for meaningful subgroup analysis. Patients aged >90 years were excluded due to shorter average life expectancy relative to the Taiwanese population(76.94 years for men and 83.74 years for women). These exclusions were necessary to ensure the scientific rigor and interpretability of the study findings. This explanation has been added to the "Materials and Methods" section of the manuscript (page 7, line 104 to 110 in tracked changes file; page 7, line 103 to 109 in without tracked changes file).

2. Some issues in the discussion need further elaboration. For example, the following finding requires more explanation: The CPLY for women peaked in the 30–59 age group ($8,308) and decreased with age, whereas for men, it peaked in the 80–89 age group ($5,758) and increased with age.

Answer:

Thank you for your insightful comments. After re-examining the data, we found that dialysis significantly contributes to total medical costs.

In males, CPLY increases with age, which aligns with trends observed in previous studies indicating that healthcare needs and related complications tend to accumulate in older age groups.

In contrast, the CPLY trend in females follows an inverse pattern compared to males. Our analysis revealed that the highest proportion of dialysis costs relative to total expenditures is observed in the 30–59 age group, accounting for 49.98% of the total. Notably, while the total hemodialysis costs for male patients amount to 27.62 million USD, female patients incur a significantly higher total of 43.52 million USD.

Since our dialysis costs are based on 'known' expenses during the current tracking period, whereas CPLY estimates future costs based on these data, the 30-59 age group—having the most documented dialysis cases—contributes to the highest projected costs. This trend aligns with the observed cost patterns across the population.

The gender disparity observed in dialysis costs underscores the importance of considering both the long-term financial burden of dialysis and the population's age structure when interpreting cost patterns. We have revised the results and discussion sections to reflect this additional context and appreciate the opportunity to clarify these points (page 16-17 , line 231 to 245 and revised Table 4 in tracked changes file; page 16 to 17, line 230 to 247 and revised Table 4 in without tracked changes file).

3. Why didn’t you divide males and females with 0IS in Table 3A? Throughout the manuscript, 0IS was not stratified by sex.

Answer:

Thank you for this observation. We did not stratify stage 0IS by sex in Table 3A because the female sample size (n=193) was relatively small compared to female in other stage. To ensure statistical reliability and avoid potential bias due to sample size imbalance, we combined both genders for analysis. Additionally, in the revised manuscript, we have clarified this point (page 13, line 192 to 197 in tracked changes file; page 13, line 191 to 195 in without tracked changes file). This approach allowed us to provide more robust and interpretable results while accounting for significant differences between groups.. This approach also highlights the notably higher mortality and CPLY observed in this subgroup, which we believe underscores the significance of this finding regardless of sex.

4. The primary driver of higher medical costs was introduced as hemodialysis. However, there is no information on how much of the costs were due to dialysis and the role of other expenditures. Providing this information would clarify it for the readers.

Answer:

Thank you for your insightful comment. We appreciate the opportunity to clarify the composition of medical expenditures, particularly the contribution of dialysis costs relative to other components.

During the follow-up period, the distribution of medical expenses varied between males and females. In females, the cost composition was as follows: medications (14.9%), laboratory tests and imaging studies (34.1%), dialysis (39.7%), and medical consumables (4.4%). In contrast, the corresponding distribution in males was medications (19.2%), laboratory tests and imaging studies (52.0%), dialysis (16.1%), and medical consumables (4.4%) (page 9, line 137 to 143 in tracked changes file; page 9, line 136 to 142 in without tracked changes file).

These findings highlight that dialysis accounted for a substantially larger proportion of total medical costs in females compared to males. This gender disparity suggests that differences in dialysis utilization may be a key factor influencing overall healthcare expenditures.

We have incorporated these details into the revised manuscript to improve clarity and address this concern.

5. In the discussion section, you mentioned, “In our subgroup analysis, which examined the interplay between cancer stage and age, we observed a notable increase in aggressive stage diagnoses among female patients aged >70 years.” More clarification is suggested regarding the interplay between cancer stage and age for each sex in the results section.

Answer:

Thank you for your insightful comment. To address your concern, we have added further clarification in the results section to elaborate on the interplay between cancer stage and age for each sex. Specifically, we have highlighted the trends observed in our subgroup analysis, where a notable increase in the proportion of aggressive stage diagnoses (MIBC) was evident among female patients aged >70 years.

Additionally, we have included a new Fig S3. to visually illustrate the trends in NMIBC, MIBC, and unknown stage distributions across different age groups for both sexes. This figure complements the text and provides a clear comparison of how cancer stage proportions change with age for males and females(page 14-15, line 211 to 214 in tracked changes file; page 14, line 210 to 213 and new Fig S3 in without tracked changes file).

Additionally, here are some minor issues suggested for improvement:

1. In the results section of the abstract, it seems that “male” and “female” have been used incorrectly in this phrase: (3.17 [0.55] years for women vs. 7.14 [0.76] years for men).

Answer:

We acknowledge that there was an error in the labeling of the data for men and women. This misprint has been corrected in the revised manuscript

2. Using MeSH terms is recommended for keywords.

Answer:

We have updated the keywords to use appropriate MeSH terms as recommended.

3. Please recheck the numbers in the tables. For example, the percentages of the number of deaths in Table 3 seem incorrect.

Answer:

We have re‐calculated the percentages in Table 3 and confirmed that the values are correct. No changes were necessary based on our recalculations.

4. In the discussion section, you mentioned, “CIS, also referred to as Tis in our study and categorized as stage 1, often…”. Doesn’t CIS categorize as stage 0?

Answer:

We agree that CIS (Carcinoma In Situ) should be categorized as stage 0 rather than stage 1. This misprint has been corrected in the revised version.

---

## [Decision Letter · Decision Letter 1]

25 Feb 2025

PONE-D-24-48193R1Gender disparities in bladder cancer: A population-based study on life expectancy and health spending in AsiaPLOS ONE

Dear Dr. Ou,

Thank you for submitting your manuscript to PLOS ONE. After careful consideration, we feel that it has merit but does not fully meet PLOS ONE’s publication criteria as it currently stands. Therefore, we invite you to submit a revised version of the manuscript that addresses the points raised during the review process.

We look forward to receiving your revised manuscript.

Kind regards,

Mazyar Zahir

Academic Editor

PLOS ONE

Reviewers' comments:

Reviewer's Responses to Questions

**Comments to the Author**

1. If the authors have adequately addressed your comments raised in a previous round of review and you feel that this manuscript is now acceptable for publication, you may indicate that here to bypass the “Comments to the Author” section, enter your conflict of interest statement in the “Confidential to Editor” section, and submit your "Accept" recommendation.

Reviewer #1: All comments have been addressed

Reviewer #2: (No Response)

Reviewer #3: (No Response)

2. Is the manuscript technically sound, and do the data support the conclusions?

Reviewer #1: No

Reviewer #2: (No Response)

Reviewer #3: (No Response)

3. Has the statistical analysis been performed appropriately and rigorously? 

Reviewer #1: Yes

Reviewer #2: (No Response)

Reviewer #3: (No Response)

4. Have the authors made all data underlying the findings in their manuscript fully available?

Reviewer #1: Yes

Reviewer #2: (No Response)

Reviewer #3: (No Response)

5. Is the manuscript presented in an intelligible fashion and written in standard English?

Reviewer #1: Yes

Reviewer #2: (No Response)

Reviewer #3: (No Response)

6. Review Comments to the Author

Reviewer #1: Dear Author

Thank you for your revision .

The items requested by me from the authors have been answered correctly and completely.

Reviewer #2: The authors have properly addressed the comments, except for one. The death percentages in Table 2 appear to be incorrect. For instance, the percentage of deaths for low-grade females is listed as 159%. Other percentages in this table also seem incorrect, such as 1238 out of 6032 being reported as 74.98%. Please re-check the numbers in this table.

Reviewer #3: - While the study uses advanced statistical techniques, it does not report p-values for the individual components and comparisons within the semiparametric method. Calculating a single p-value for the entire extrapolation process is challenging due to the complexity and multi-step nature of the method. However, it is recommended to report p-values for individual components, such as the comparison of survival rates and covariates using Cox proportional hazards models, to provide a clearer understanding of the statistical significance of the findings.

- The method assumes homogeneity in the patient data concerning the variables incorporated within the model. Variability in treatment, access to healthcare, and other patient-specific factors may not be comprehensively considered and ought to be recognized as a limitation.

- Extending the survival curve month by month involves extrapolating beyond the observed data, introducing uncertainties. This approach assumes that the trends observed in the initial months will continue, which might not always be the case. Including sensitivity analyses or validating the model with external data could strengthen the findings.

- The reference group created using the Monte Carlo method based on life tables may not fully capture all variables affecting the general population's survival, such as comorbidities or socio-economic factors. This could lead to biased comparisons and should be discussed as a potential limitation.

7. PLOS authors have the option to publish the peer review history of their article (what does this mean? ). If published, this will include your full peer review and any attached files.

**Do you want your identity to be public for this peer review?** For information about this choice, including consent withdrawal, please see our Privacy Policy .

Reviewer #1: No

Reviewer #2: No

Reviewer #3: No

---

## [Author Response · Author response to Decision Letter 2]

17 Mar 2025

Reviewer #1:

Dear Author

Thank you for your revision. The items requested by me from the authors have been answered correctly and completely.

Answer:

We sincerely appreciate your time and effort in reviewing our manuscript. We are grateful for your positive feedback and for acknowledging that we have adequately addressed your previous comments. As per your assessment, no additional revisions were required. Nonetheless, we have carefully re-reviewed the entire manuscript to ensure clarity, accuracy, and completeness, and we have made further refinements based on feedback from the other reviewers to enhance the overall quality of our work.

Thank you again for your constructive review and support.

Reviewer #2:

The authors have properly addressed the comments, except for one. The death percentages in Table 2 appear to be incorrect. For instance, the percentage of deaths for low-grade females is listed as 159%. Other percentages in this table also seem incorrect, such as 1238 out of 6032 being reported as 74.98%. Please re-check the numbers in this table.

Answer:

We appreciate your careful review and for highlighting the discrepancies in Table 2. Upon re-examining our calculations, we discovered that errors arose from a misalignment in data extraction and an incorrect denominator for percentage calculations. We have thoroughly revised Table 2 (pages 12–13, revised Table 2) so that all reported values accurately reflect the proportion of deaths in each cohort. The corrected percentages now fall within expected ranges. Additionally, we reviewed all other tables and data sources to ensure consistency and accuracy throughout the manuscript.

Thank you again for your constructive feedback, which has significantly improved the precision of our study.

Reviewer #3:

1. While the study uses advanced statistical techniques, it does not report p-values for the individual components and comparisons within the semiparametric method. Calculating a single p-value for the entire extrapolation process is challenging due to the complexity and multi-step nature of the method. However, it is recommended to report p-values for individual components, such as the comparison of survival rates and covariates using Cox proportional hazards models, to provide a clearer understanding of the statistical significance of the findings.

Answer:

Thank you for your valuable feedback. In response, we have performed a multivariable Cox regression analysis including sex, age, pathological grade, and cancer stage (pages 8, lines 119–123). This analysis provides hazard ratios with 95% confidence intervals and p-values for each variable—for example, male sex is associated with a lower risk of death (HR 0.90, 95% CI 0.87–0.94, p < 0.01), while increasing age, high-grade tumors, and advanced-stage disease significantly elevate mortality risk (all p < 0.01). These results are detailed in the newly added Supplement Table S5 and summarized in the Results section (pages 16, lines 226–232, and new S2 Table). We believe this additional analysis enhances the statistical rigor and clarity of our findings.

2. The method assumes homogeneity in the patient data concerning the variables incorporated within the model. Variability in treatment, access to healthcare, and other patient-specific factors may not be comprehensively considered and ought to be recognized as a limitation.

Answer:

Thank you for this important comment. We acknowledge that our model assumes patient homogeneity for the included variables. In reality, variations in treatment, healthcare access, and other patient-specific factors may introduce residual confounding. We have revised the Limitations section to include the following statement (page 29, lines 458–460):

"Moreover, our model assumes patient homogeneity for the included variables. In reality, variations in treatment, healthcare access, and other factors may introduce residual confounding."

We believe this revision clearly recognizes the inherent heterogeneity in the patient population and enhances the transparency of our study.

3. Extending the survival curve month by month involves extrapolating beyond the observed data, introducing uncertainties. This approach assumes that the trends observed in the initial months will continue, which might not always be the case. Including sensitivity analyses or validating the model with external data could strengthen the findings.

Answer:

Thank you for your insightful comment. In response, we have inserted a sentence in the Methods section (page 8, lines 117–119; page 16, lines 221–226) describing a sensitivity analysis in which we compared life expectancy estimates extrapolated from the first 6 years of follow-up with 12-year Kaplan–Meier estimates (new Supplement S1 Table). The analysis revealed minimal relative bias, thereby supporting the robustness of our extrapolation approach despite inherent uncertainties. We hope this revision adequately addresses your concern.

4. The reference group created using the Monte Carlo method based on life tables may not fully capture all variables affecting the general population's survival, such as comorbidities or socio-economic factors. This could lead to biased comparisons and should be discussed as a potential limitation.

Answer:

Thank you for your valuable comment regarding our reference group. We recognize that using a Monte Carlo simulation based on life tables may not capture key factors—such as comorbidities and socioeconomic status—that affect the general population’s survival, potentially biasing comparisons. In response, we have revised the Limitations section (page 29, lines 457–458) to include the following statement:

"Additionally, the Monte Carlo-based reference group may not capture key factors (e.g., comorbidities, socioeconomic status), potentially biasing comparisons."

We believe this addition enhances the transparency of our study by clearly acknowledging this limitation.

---

## [Decision Letter · Decision Letter 2]

15 Apr 2025

Gender disparities in bladder cancer: A population-based study on life expectancy and health spending in Asia

PONE-D-24-48193R2

Dear Dr. Ou,

We’re pleased to inform you that your manuscript has been judged scientifically suitable for publication and will be formally accepted for publication once it meets all outstanding technical requirements.

Kind regards,

Mazyar Zahir, MD

Academic Editor

PLOS ONE

Additional Editor Comments (optional):

Reviewers' comments:

Reviewer's Responses to Questions

**Comments to the Author**

1. If the authors have adequately addressed your comments raised in a previous round of review and you feel that this manuscript is now acceptable for publication, you may indicate that here to bypass the “Comments to the Author” section, enter your conflict of interest statement in the “Confidential to Editor” section, and submit your "Accept" recommendation.

Reviewer #2: All comments have been addressed

Reviewer #3: All comments have been addressed

2. Is the manuscript technically sound, and do the data support the conclusions?

Reviewer #2: (No Response)

Reviewer #3: Yes

3. Has the statistical analysis been performed appropriately and rigorously? 

Reviewer #2: (No Response)

Reviewer #3: Yes

4. Have the authors made all data underlying the findings in their manuscript fully available?

Reviewer #2: (No Response)

Reviewer #3: (No Response)

5. Is the manuscript presented in an intelligible fashion and written in standard English?

Reviewer #2: (No Response)

Reviewer #3: (No Response)

6. Review Comments to the Author

Reviewer #2: (No Response)

Reviewer #3: The items I previously requested from the authors have been addressed in a timely manner. I appreciate their responsiveness and the clarity of their answers.

7. PLOS authors have the option to publish the peer review history of their article (what does this mean? ). If published, this will include your full peer review and any attached files.

**Do you want your identity to be public for this peer review?** For information about this choice, including consent withdrawal, please see our Privacy Policy .

Reviewer #2: No

Reviewer #3: **Yes: ** Nasrin Borumandnia

---

## [Editor Report · Acceptance letter]

PONE-D-24-48193R2

PLOS ONE

Dear Dr. Ou,

I'm pleased to inform you that your manuscript has been deemed suitable for publication in PLOS ONE. Congratulations! Your manuscript is now being handed over to our production team.

Kind regards,

on behalf of

Dr. Mazyar Zahir

Academic Editor

PLOS ONE